# Dataless Knowledge Fusion by Merging Weights of Language Models

**Xisen Jin[§], Xiang Ren[§], Daniel Preoţiuc-Pietro[†], Pengxiang Cheng[†]**

[§]University of Southern California      [†]Bloomberg
`{xisenjin, xiangren}@usc.edu`
`{dpreotiucpie, pcheng134}@bloomberg.net`

## Abstract

Fine-tuning pre-trained language models has become the prevalent paradigm for building downstream NLP models. Oftentimes fine-tuned models are readily available but their training data is not, due to data privacy or intellectual property concerns. This creates a barrier to fusing knowledge across individual models to yield a better single model. In this paper, we study the problem of merging individual models built on different training data sets to obtain a single model that performs well both across all data set domains and can generalize on out-of-domain data. We propose a dataless knowledge fusion method that merges models in their parameter space, guided by weights that minimize prediction differences between the merged model and the individual models. Over a battery of evaluation settings, we show that the proposed method significantly outperforms baselines such as Fisher-weighted averaging or model ensembling. Further, we find that our method is a promising alternative to multi-task learning that can preserve or sometimes improve over the individual models without access to the training data. Finally, model merging is more efficient than training a multi-task model, thus making it applicable to a wider set of scenarios.[1]

## 1 Introduction

The dominant paradigm for solving NLP tasks ranging from classification to sequence tagging involves fine-tuning a pretrained language model (PLM) using task-specific labeled data (Devlin et al., 2019; He et al., 2021). This results in specialized models that are explicitly trained to run inference over a single domain and task. Multi-task learning has shown that leveraging information across domains or tasks can be beneficial if the data sets, data set size and algorithms are well selected (Phang et al., 2018; Pruksachatkun et al., 2020; Poth et al., 2021; Weller et al., 2022). Combining knowledge of multiple data sets in a single model can lead to better overall performance on in-domain data (Poth et al., 2021), can better generalize on out-of-domain data (Wang et al., 2020b) and results in a model that is more practical and parameter efficient than maintaining specialized models.

However, the multi-task learning setup suffers from two practical limitations. First, the training process requires access to the original labeled data, which may not be realistic as annotated data may be private to the agent fine-tuning the model which can happen in order to ensure data or annotation privacy or to guard intellectual property to annotations. Second, because a significant amount of data or task combinations are not beneficial to performance (Poth et al., 2021), building a single model requires training on all data set combinations to identify the optimal one, which can be prohibitive especially if there are many available source data sets or models.

Model merging is defined as combining multiple models into a single one in parameter space without access to data (Matena & Raffel, 2021). This technique provides an alternative to building a single model while satisfying data privacy constraints. Weight merging algorithms usually also have a closed-form solution, making them very efficient as no retraining is necessary, thus enabling usage even when a large number of data sets or model combinations are available. Merging can be considered as an alternative to model ensembling (Opitz & Maclin, 1999; Rokach, 2010), where the

---

[1]The code is available at: `https://github.com/bloomberg/dataless-model-merging`

Figure 1: Diagram containing the problem formation for model merging and its comparison to other setups including multi-task learning, model ensembling and federated learning. Models $f_{1..N}$ trained by individuals or organizations are released to the user (optionally with some statistics) but the training data $D_{1..N}$ is kept private.

outputs of individual models are combined to produce the final prediction. Model merging algorithms are a key step in federated learning (McMahan et al., 2017; Lin et al., 2022), where multiple agents train their own model using private data and share only model updates with other models. However, in federated learning, model merging happens in multiple rounds of updates, after which the merged model is broadcast to all agents before the next round of training with private data. This *dataless* model merging is thus an extreme case of federated learning, where a single round of synchronization is admissible. Figure 1 provides an overview of the various related setups.

We thus aim to use model merging to build a single model that can be used for inference on multiple domains or tasks and can generalize to new domains, in line with Wang et al. (2020b). In contrast, simple averaging of weights for model merging was used by existing works such as Wortsman et al. (2022) to improve the performance of a specific model, where weight averaging was done over models fine-tuned using the same data set with different hyperparameters. Separately, Matena & Raffel (2021) focus on improving performance over a single target task by leveraging models trained on other *donor* tasks by merging models using Fisher-weighted averaging.

This paper focuses on merging fine-tuned models that originate from pre-trained language models with the *same architecture* and *pretrained weights*. We introduce a novel model merging method named Regression Mean (RegMean), which is computationally efficient and extendable to merging any number of models. The method is inspired by the optimal solution for linear models that minimizes $\ell^2$ distance between merged and individual models and has a closed form solution. We evaluate model merging algorithms in setups that range in complexity and type of fused knowledge. The experimental results across multiple model types (e.g. RoBERTa, T5, DeBERTa) show that our proposed method consistently and significantly outperforms other model merging and ensembling baselines and achieves higher generalization performance than the best individual models on out-of-domain data sets across several data collections.

Our **contributions** are three-fold: (1) A novel model merging algorithm (Regression Mean); (2) an evaluation protocol for model merging algorithms that tests both in-domain and out-of-domain generalization ability; (3) analysis of computation and parameter efficiency across setups.

## 2 DATALESS MODEL MERGING FOR KNOWLEDGE FUSION

We consider the problem formulation that there are two main roles in the framework: (1) the agents (*e.g.*, individuals or organizations) that train and release models; (2) the developers who aim to build a single model by fusing knowledge of multiple available models. Each agent $i \in \{1..N\}$ fine-tunes a language model (LM) $f_i$ of pre-trained weights $\theta_{\mathrm{LM}}$ over their private labeled dataset $D_i = \langle X_i, Y_i \rangle$ to obtain fine-tuned model weights $\theta_i$, where $X_i \in \mathbb{R}^{N_i,*}$ are inputs, $Y_i \in \mathbb{R}^{N_i,*}$ are labels and $N_i$ is the number of annotated examples. The agents keep the labeled data set $D_i$ private. In addition to the fine-tune model weights $f_i(\cdot; \theta_i)$, the agents can also optionally disseminate certain statistics $S_i$, as long as these do not leak information about the labeled data set $D_i$.

In turn, the developers use the fine-tuned model weights $f_i(\cdot; \theta_i)$ and statistics $S_i$ as inputs to a merging function $g$. The merging function is applied to a subset of fine tuned models $\mathcal{K} \subseteq \{1..N\}$ (of size $K = |\mathcal{K}|$) to obtain parameters $\theta_{M_\mathcal{K}}$ of a merged model $f_{M_\mathcal{K}}$, where $\theta_{M_\mathcal{K}} = g(\theta_\mathcal{K}, S_\mathcal{K})$. In general, we expect the function $g$ to be computationally efficient and to produce $\theta_{M_\mathcal{K}}$ with a closed-form formulation.

Figure 2: Comparison between Simple, Fisher, and RegMean for merging transformer-based language models. Fisher and RegMean require Fisher Information matrix or inner product matrices of layer inputs, but neither of them requires training data. For linear models, RegMean produces optimal weights that minimize $\ell^2$-distance to individual model predictions on the corresponding training sets.

# 3    REGRESSION MEAN FOR MODEL MERGING

The key role in the model merging setup is played by the merging function $g$. We start with briefly introducing existing techniques for model merging, followed by the basic intuition for our proposed method, which we then extend to transformer-based language models. The underlying **assumption** is that the model architecture for all models $f_i$ is the same, allowing for element-wise operations if needed and resulting in a merged model $f_{M_\mathcal{K}}$ of the same architecture and size as any individual model. We also assume models are fine-tuned from the same pretrained LM checkpoint. The study of methods that relax this constraint are outside the scope of this paper and are left for future work.

## 3.1    PRELIMINARIES

**Simple Averaging (Simple)**  computes the merged weights as the element-wise arithmetic mean of the weights of all other models: $\theta_{M_\mathcal{K}} = 1/K \sum_i^{i \in \mathcal{K}} \theta_i$. This technique was proved to be effective when merging model weights that are already similar or in a similar space, such as checkpoints generated after each epoch in a training process (Wortsman et al., 2022). We expect simple averaging to under-perform when model weights live in a different space and are substantially different to each other, such as when merging models trained with different data or when performing merging for models fine-tuned after the entire training process, as opposed to synchronizing models after rounds as in the federated learning setup.

**Fisher-Weighted Averaging (Fisher)**   aims to address the limitation of simple averaging of weights with potentially different importance. The method relies on computing per-weight importance $F_i$ for each individual model $i$, and reweighting the weights with this importance factor during merging as follows: $\theta_{M_\mathcal{K}} = \sum_i^{i \in \mathcal{K}} F_i \theta_i / \sum_i^{i \in \mathcal{K}} F_i$. Here, $F_i$ is the diagonal of the Fisher Information matrix, where $F_i = \mathbb{E}_{x \sim D_i} \mathbb{E}_{y \sim p_\theta(y|x)} \nabla_{\theta_i} (\log p_{\theta_i}(y|x_i))^2$. Intuitively, $F_i$ measures averaged gradient norm of parameters w.r.t. log likelihood of each label, where parameters with high average norms are considered important.

## 3.2    MERGING LINEAR MODELS

Next, we recast the problem of model merging as a straightforward optimization problem. We start by inferring the optimal solution of merging two linear regression models trained on different data distributions and analyze its relationship to Simple averaging.

Consider two linear models $f_1(x) = W_1^T x$ and $f_2(x) = W_2^T x$, where $x \in \mathbb{R}^m$, and $W_1, W_2 \in \mathbb{R}^{m \times n}$, that are trained on two different annotated datasets $\langle X_1, y_1 \rangle, \langle X_2, y_2 \rangle$, where $X_1 \in \mathbb{R}^{N_1 \times m}$ and $X_2 \in \mathbb{R}^{N_2 \times m}$. Each row in $X_i$ corresponds to a training example. Our goal is to obtain a single merged model $f_M(x) = W_M^T x$ with outputs similar to $f_1$ on $X_1$ and $f_2$ on $X_2$. With $\ell^2$ distance as the metric, the optimization problem can be formulated as:

$$\min_W \quad \|W^T X_1 - W_1^T X_1\|^2 + \|W^T X_2 - W_2^T X_2\|^2. \tag{1}$$

Eq. 1 describes a linear regression problem, where the inputs are $[X_1; X_2]$ (row concatenation of $X_1$ and $X_2$) and the targets are $[W_1^T X_1; W_2^T X_2]$, which has a closed form solution $W_M = (X_1^T X_1 + X_2^T X_2)^{-1}(X_1^T X_1 W_1 + X_2^T X_2 W_2)$. The algorithm extends to merging $K$ models $W_i, i \in \mathcal{K}$ with little modifications to the optimization problem in Eq. 1:

$$W_M = (\sum_i^{i \in \mathcal{K}} X_i^T X_i)^{-1} \sum_i^{i \in \mathcal{K}} (X_i^T X_i W_i). \tag{2}$$

We refer to Eq. 2 as Regression Mean (RegMean). To summarize, to merge a linear model $f_i$ with other models, we pre-compute the inner product matrices of training data $X_i^T X_i$; we do not recompute $X_i^T X_i$ when merging with different models. The merger retrieves the weights and inner product matrices of inputs of individual models and compute the weights as in Eq. 2.

**Interpretation**. RegMean can be also interpreted as reweighting and linearly combing rows in weight matrices, where the diagonal items of $X_i^T X_i$ mainly reweight the rows, while non-diagonal items linearly combine them. In an extreme case when $X_i^T X_i$ is diagonal, RegMean simply reweights the rows in $W_i$ by the importance of neurons. Besides, when all $X_i^T X_i$ (or all $X_i$) are the same, Eq. 2 transforms into simple averaging, *i.e.*, $W_M = 1/K \sum_i^{i \in \mathcal{K}} W_i$.

## 3.3 REGMEAN FOR TRANSFORMER LANGUAGE MODELS

Transformer models consist of feed forward layers and attention heads where linear layers are important components. For all linear layers, we independently apply RegMean. We record $X_i^{(j)T} X_i^{(j)}$ of each linear layer $f^{(j)}$, where $X_i^{(j)}$ is the input features of the linear layer. The other types of weights, such as embeddings and bias terms, that represent a small portion of the overall parameter set are merged using simple averaging.

**Reducing Non-Diagonal Items of Inner Product Matrices.** We empirically find that directly applying Eq. 2 for merging yields degenerated models in case of some pre-trained LM architectures. We therefore decrease the non-diagonal items of the inner product matrices by multiplying them with a scalar $\alpha$ (set as 0.9 most of the times). This also corresponds to adding a regularization term in the optimization objective in Eq. 1 that penalizes the Euclidean distance between the merged weights $W_M$ and individual model weights $W_{1..K}$.

We include a formal derivation and proof in Appendix A. We illustrate RegMean in Figure 2 and summarize the complete RegMean method in Algorithm 1.

---

**Algorithm 1:** RegMean for Transformer Language Models

---

**Data:** Individual Models $f_{1..K}$, Number of linear layers $J$, inner product matrices $G_i^{(j)} = X_i^{(j)T} X_i^{(j)}$ for all linear layers $1 \le j \le J$ and models $1 \le i \le K$, Scaling factor of non-diagonal items $\alpha$
**Result:** Merged model $f_M$
**for** $j$ **in** $1, 2, ..., J$ **do**
 $W_1^{(j)}, W_2^{(j)}..., W_K^{(j)} \leftarrow$ getLinearWeights$(f_{1..K}, j)$ ;
 Reduce non-diagonal items of inner product matrices $G_i^{(j)}$ as $\tilde{G}_i^{(j)} \leftarrow \alpha G_i^{(j)} + (1 - \alpha)\text{diag}(G_i^{(j)})$ ;
 $W_M^{(j)} \leftarrow (\sum_i^{i \in \mathcal{K}} \tilde{G}_i^{(j)})^{-1} \sum_i^{i \in \mathcal{K}} (\tilde{G}_i^{(j)} W_i^{(j)})$ and set the weight as $W_M^{(j)}$ in $f_M$
**end**
Average weights as $W_M = \frac{1}{K} \sum_i^{i \in \mathcal{K}} W_i$ for weights other than linear layer weights in $f_M$

---

## 3.4 PROPERTIES OF REGMEAN

**Computational Efficiency.** Inner product matrices of all linear layer inputs can be computed within one single forward pass over training data after individual models are trained. It is more efficient than computing Fisher Information matrices, which requires an additional backward pass to compute gradients.

**Memory Overhead**. The memory overhead of inner product matrices is $\sum_{j=1}^{J} d_j^2$, where $J$ is the number of linear layers in the model and $d_j$ is the input dimension of linear layers. For transformer models, this overhead is comparable to the number of parameters and Fisher Information matrices.

**Data Privacy**. It should be noted that RegMean never requires training data $X_i$ when merging; instead, it only requires low-dimensional inner product matrices. The agents that release the models can share the matrices without sharing the private training data and their labels.

## 4 EXPERIMENTAL SETUP

### 4.1 EVALUATION SETTINGS

We expect two major benefits of merging models for the developer. First, by combing knowledge of individual models $f_{1..N}$ (or a subset $\mathcal{K}$ of them, $f_{\mathcal{K}}$) trained on $D_{1..N}$, we expect the resulting

merged model $f_M$ to achieve competitive test performance across all datasets $D_{1..N}$. This model is useful for example when the test distribution is a mixture of $D_{1..N}$. In addition, a single model has the additional advantage of being able to run inference across multiple domains when the user of the model provides data from one of the domains, but is not aware of the domain label (Wang et al., 2020b). In our case, $D_{1..N}$ can represent different non-i.i.d. partitions of the same dataset, different domains for the same task or different tasks altogether.

Second, we expect the merged model to achieve higher *out-of-domain (OOD)* generalization ability. Formally, we evaluate the performance of the merged model $f_M$ over the out-of-domain test sets $D_{1..N_o}^o$ where the data distributions are different from any of $D_{1..N}$.

**Datasets**. We use the GLUE datasets (Wang et al., 2018) for studying merging models trained for non-i.i.d. partitions and merging models trained for different tasks. We use emotion classification and named entity recognition (NER) as base tasks for studying merging models trained on different domains of the same task. For emotion classification, we use the collection of preprocessed datasets from (Oberländer & Klinger, 2018). We choose 5 high-resource datasets for training individual models and 5 low-resources datasets for evaluation of out-of-domain generalization ability. For NER tasks, we use 6 domains in OntoNotes (Hovy et al., 2006) for training individual models, and use CoNLL (Sang & De Meulder, 2003) and Twitter NER (Rijhwani & Preotiuc-Pietro, 2020) to measure out-of-domain generalization performance. We include details of datasets in Apppendix B.

**Metrics**. In the case of merging models trained on non-i.i.d. partitions of the same dataset, we evaluate the merged models over a single test set with a joint distribution of all partitions. For merging models trained on different domains or tasks, we measure the performance over all single domains or tasks incorporated into merging and take their macro-average. For out-of-domain evaluation, we similarly take macro-average over the performance over the out-of-domain test sets.

## 4.2 COMPARED METHODS

**Model Merging.** For model merging algorithms, we compare the performance of RegMean with the previously introduced methods of simple averaging (**Simple**) (Wortsman et al., 2022) and Fisher-weighted averaging (**Fisher**) (Matena & Raffel, 2021).

**Model Ensembling.** Model ensembling represents an alternative to model merging when access to the original data is not available. We thus build an ensemble model (**Ensemble**) by obtaining all logits from the individual model predictions and averaging them before doing an argmax.

**Individual Models.** To provide context into the benefits of merging, we report the performance of individual models involved in merging. We thus report: (1) the average performance of all individual models (**Avg.** $f_{1..N}$); (2) the performance of the best *single* individual model (**Best.** $f_{1..N}$), as determined by using the validation set; (3) the performance of the individual models corresponding to the training data set for each test set (**Domain-Specific**).

**Multi-task Learning (MTL).** We also consider MTL which trains a single model over the joint training data sets $D_{1..N}$. We note that the multi-task method should represent an upper-bound for model merging, as multi-task learning has access to the original labeled data which it can leverage to train a better model when compared to dataless approaches such as model merging. Depending on the data sets, the task can be the same (*e.g.*, emotion prediction) or different (*e.g.*, GLUE tasks).

## 4.3 EXPERIMENT DETAILS

**Pre-trained Models**. We initialize all models $f_i$ using the same architecture and by using the same pre-trained model weights $\theta_{LM}$. We experiment with multiple pre-trained models as starting points for merging. We experiment with both encoder-only models including the classic RoBERTa-base (Liu et al., 2019) and state-of-the-art models like DeBERTa-large-v3 (He et al., 2021) and with encoder-decoder models represented by T5-base-v1.1 (Raffel et al., 2020). We note that T5-base-v1.1 is not applicable to sequence labelling tasks represented by our NER experiments. Further training details are in Appendix B.

**Model Initialization.** It has been shown that model merging is more successful when individual models share the same weight initialization (McMahan et al., 2017). In this paper, we focus on merging fine-tuned language models of the same architectures and initialized from the same pre-trained model weights $\theta_{LM}$ before fine-tuning. For new classification heads, we present the results of both shared initialization (**Same Head Init, SH**) and different initialization (**Diff Head Init,**

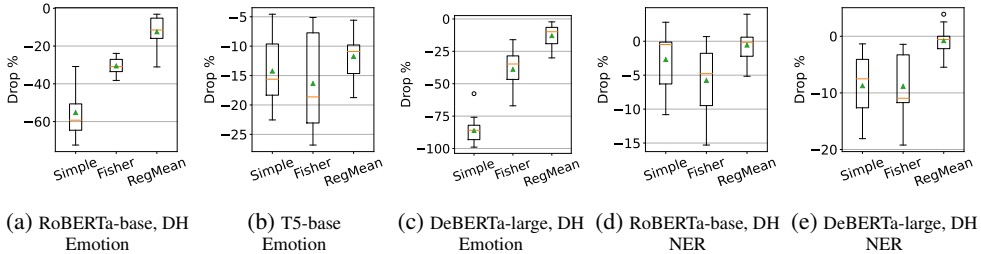

Figure 3: Relative performance drop (%) of pairwise merged models compared to the **domain-specific** models. Positive values indicate performance improvement after merging. The boxplots summarize results over 10 ($\mathcal{C}_5^2$) or 15 ($\mathcal{C}_6^2$) combinations of 5 or 6 domain-specific models in Emotion and NER. The triangles denote the mean. Note that y-axes are not in the same scale.

**DH**), as our proposed method is amenable to both. This does not apply to T5 where we fine-tune the pretrained LM head for prediction.

**Hyperparameters.** We set the non-diagonal multiplier $\alpha$ in RegMean to $0.9$, with the exception of T5-base models, where it is $0.1$. We compute inner product matrices with at most $1,000$ training batches. Sensitivity analysis of hyperparameters is presented in Section 5.3 and Appendix C.

## 5 RESULTS

The main goal of our experiments is to benchmark the performance of different dataless model merging methods and compare these with individual model performance before merging. In addition, we aim to situate these methods in context of other methods which represent upper bounds due to having access to more information (*i.e.* data for fine-tuning) than model merging.

Our experiments examine knowledge fusion from two perspectives: (1) in-domain performance over test data sets similar to those over which individual models are trained, and (2) out-of-domain generalization performance over data sets from held-out domains or tasks. We study performance dynamics in a range of scenarios ranging in difficulty. First, we study a simple scenario where merging is performed on models are trained on non-i.i.d. partitions of the same data set. Next, we study merging of models trained on different domains of the same task and lastly merging models trained on different tasks.

### 5.1 MODEL MERGING FOR FUSING IN-DOMAIN KNOWLEDGE

**Merging Models Trained on Non-i.i.d. Partitions.** We start with a setup in which we merge models trained on non-i.i.d. partitions of the same data set, which is simulated using synthetic data splits over the 8 tasks in the GLUE benchmark. For each task, we split training data into two partitions with 1,000 training examples with different label distributions (details in Appendix B). We then fine-tune 8 pairs of individual models over the two partitions and merge each pair of the models. The merged models are evaluated on the official validation sets (*i.e.*

|       | Avg. $f_{1..N}$ | Simple | Fisher | RegMean |
|-------|------|--------|--------|---------|
| SST-2 | 86.80 | 89.98 | 90.00 | **90.23** |
| MRPC  | 79.34 | 80.44 | 80.39 | **81.96** |
| STS-B | 87.50 | 87.86 | 88.15 | **88.20** |
| ... |  |  |  |  |
| 8-task Avg. | 71.76 | 74.22 | 75.25 | **75.27** |

Table 1: Merging models trained on Non-i.i.d. partitions of GLUE tasks. We compare the performance of the merged models (Simple, Fisher, RegMean) and the average performance of each pair of individual models (Avg. $f_{1..N}$) over the joint validation sets.

with a joint distribution of both partitions). In Table 1, we find that model merging consistently improves over average performance of individual models across the 8 tasks. This verifies that weight merging allows combining knowledge from individual models and can lead to a more powerful single model. We further note that RegMean outperforms simple averaging and is similar in performance to Fisher-weighted averaging. This is a proof-of-concept that model merging and RegMean work in a simple scenario.

**Merging Models Trained on Different Domains.** We next shift to a more challenging setup where individual models are trained on data from different domains of the same task.

*Pairwise Merging.* We start by merging pairs of models trained on different domains. For emotion classification and NER, we have 10 ($\mathcal{C}_5^2$) and 15 ($\mathcal{C}_6^2$) combinations of domain-specific mod-

| Model (→) Method (↓) | Emotion | | | NER | |
| --- | --- | --- | --- | --- | --- |
| | RoBERTa-base Same / Diff Head Init. | T5-base | DeBERTa-large Same / Diff Head Init. | RoBERTa-base | DeBERTa-large |
| Avg. $f_{1..N}$ | 18.91 | 32.16 | 27.56 | 77.02 | 76.69 |
| Best. $f_{1..N}$ | 23.98 | 34.19 | 33.86 | 88.46 | 84.82 |
| Ensemble | 27.21 / 26.82 | 38.89 | 28.93 / 28.44 | 85.45 | 86.40 |
| Simple | 21.31 / 0.00 | 39.52 | 2.96 / 0.00 | 81.63 | 55.37 |
| Fisher | 28.27 / 24.36 | 39.28 | 10.88 / **20.16** | 76.75 | 51.01 |
| RegMean | **38.73 / 32.56** | **40.32** | **38.31** / 18.83 | **85.68** | **85.51** |
| Domain-Specific | 51.02 | 49.38 | 52.53 | 88.61 | 88.31 |
| MTL | 47.75 | 49.06 | 51.52 | 90.41 | 90.12 |

Table 2: **In-domain performance** when merging all 5 emotion classification models or 6 NER models. Simple, Fisher and RegMean are the model merging algorithms for comparison. **Bold** numbers indicate the best performance across different model merging algorithms.

els respectively. The boxplots in Fig. 3 summarize the relative performance drop compared to domain-specific models as $\frac{1}{N(N-1)}\sum_{i=1}^{N}\sum_{j=1,j\neq i}^{N}[\mathcal{M}(f_{M_{i,j}}, D_i) - \mathcal{M}(f_i, D_i)]/\mathcal{M}(f_i, D_i)$, where $\mathcal{M}(f, D)$ denotes the metric score obtained by evaluating $f$ on the test set of $D$. The performance drop is reasonable as the merged model can run inference on both domains; when the test set is a mixture of all domains, the merged model usually outperforms single individual models, as we will see in the next paragraph. We see clear differences between model merging algorithms, where RegMean performs the best. On RoBERTa-base and DeBERTa-large, RegMean reduces performance drop on Emotion from 55% to 12% and 85% to 15% compared to simple average.

*Merging All Domain-Specific Models.* We further experiment in a setup of merging all 5 or 6 domain-specific models on Emotion Classification and NER. Table 2 summarizes the results. Results show that merging all models is a challenging setup. The large differences between the average and the best performance of individual models (Avg. $f_{1..N}$ and Best $f_{1..N}$) indicate the performance of individual models have a high variance. As a result, model ensembling suffers from poor individual models: the improvements are mostly marginal compared to Best $f_{1..N}$, while on DeBERTa-large on Emotion, the performance is actually lower. In contrast, MTL improves performance significantly over Best $f_{1..N}$ and achieves performance similar to or better than domain-specific models, which implies a single model is capable of encoding knowledge of all domains in our setup.

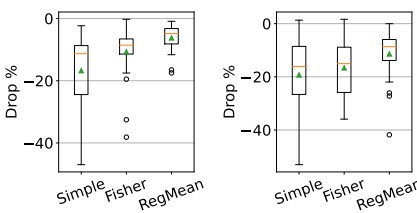

(a) DistilBERT-base    (b) RoBERTa-base

Figure 4: Relative performance drop (%) of merged models compared to task-specific models in our pairwise model merging experiments over GLUE.

We then compare three different merging algorithms. RegMean achieves the best in-domain performance on both Emotion and NER tasks, except for DeBERTa-large on Emotion, where Fisher performs slightly better. Simple averaging performs poorly (except for T5), especially on RoBERTa-base and DeBERTa-large in the emotion tasks. We note that Fisher clearly under-performs RegMean in our previous pairwise merging experiments; Fisher-weighted averaging may actually produce a merged model that is very similar to one of the individual model. RegMean also outperforms ensembling in all but one of the five scenarios.

RegMean also clearly outperforms Best $f_{1..N}$ on RoBERTa and T5-base on Emotion, which makes model merging with RegMean useful for performance purposes, in addition to the practical convenience of deploying and maintaining a single model for multiple domains.

**Merging Models Trained on Different Tasks.** We also experiment with merging models trained on different tasks using DistilBERT-base and RoBERTa-base. We train individual models with full training data of 8 GLUE tasks. We do not merge task-specific classification heads as these can have different dimensions depending on the task and output space. We summarize the results in Figure 4. We again see a similar pattern when comparing model merging techniques with RegMean clearly improving over Simple averaging and Fisher-weighted averaging.

## 5.2 MODEL MERGING FOR OUT-OF-DOMAIN GENERALIZATION

| Model ($\rightarrow$) | Emotion-Heldout | | | NER-CoNLL | | NER-Twitter | |
|---|---|---|---|---|---|---|---|
| Method ($\downarrow$) | RoBERTa-base Same / Diff Head Init. | T5 base | DeBERTa-large Same / Diff Head Init. | RoBERTa base | DeBERTa large | RoBERTa base | DeBERTa large |
| Avg. $f_{1..N}$ | 21.71 | 30.30 | 20.76 | 67.91 | 69.76 | 44.50 | 41.10 |
| Best. $f_{1..N}$ | 30.06 | 37.54 | 31.10 | 80.24 | 83.33 | 58.40 | 52.48 |
| Ensemble | 11.92 / 10.90 | 28.10 | 12.55 / 10.65 | 85.45 | 80.58 | 48.43 | 48.59 |
| Simple | 11.17 / 0.00 | 38.77 | 0.81 / 0.00 | 73.92 | 54.95 | 48.07 | 29.93 |
| Fisher | 20.67 / **18.76** | 37.84 | 5.80 / **32.04** | 68.68 | 42.13 | 45.81 | 26.05 |
| RegMean | **22.75** / 15.53 | **39.58** | **16.40** / 5.02 | **78.27** | **80.43** | **50.70** | **46.76** |
| MTL | 28.29 | 37.71 | 30.94 | 78.53 | 80.60 | 33.21 | 43.55 |

Table 3: **Out-of-domain performance** when merging all 5 emotion classification models or 6 NER models. **Bold** numbers indicate the best performance across different model merging algorithms.

**Out-of-Domain Generalization when Merging all Domain-Specific Models.** Table 3 summarizes OOD generalization performance when merging all domain-specific models. We see a similar pattern in OOD generalization performance where RegMean in general performs the best across all model merging algorithms. The performance is lower than Fisher only on RoBERTa-base and DeBERTa-large with different head initialization. We also see that RegMean outperforms model ensembling in most cases, which is comparable in the amount of information it can use. Further, on the emotion classification data sets, it is notable that RegMean achieves higher OOD performance than the best $f_{1..N}$ on T5-base. We also found that knowledge fusion itself can negatively impact performance when there are poor individual models: on NER, all merging algorithms and even MTL does not achieve better OOD performance on CoNLL and Twitter than picking the Best $f_{1..N}$, as previously indicated in Wang et al. (2020b).

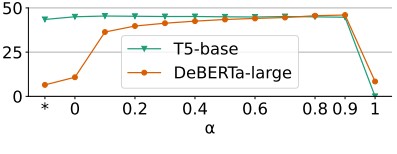

(a) Merging two models

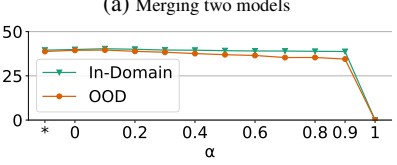

(b) Merging all models, T5-base

Figure 5: Performance of RegMean with different values of $\alpha$ in Emotion Classification. $*$ denotes for Simple Average.

**Incrementally Merging a Subset of Models**. In a scenario where OOD performance of each individual model is known (*e.g.*, when the validation sets of the OOD data sets are provided), we can mitigate the impact of having poor individual models by merging only a subset $\mathcal{K} \subseteq \{1..N\}$ of models. We apply a similar technique as Wortsman et al. (2022); Ramé et al. (2022) which greedily identifies new individual models to merge. We use their OOD performance on the validation sets to incrementally add models and plot the results in Figure 6. In general, merging only a subset of models is better than merging all models, *e.g.*, on RoBERTa-base with the same head initialization, RegMean outperforms Best $f_{1..N}$ by merging only two models.

## 5.3 DISCUSSION

**Pre-trained Model Impact in Merging.** Our results also show that the underlying pre-trained model is an important factor that affects the performance of merged models. Overall, merging T5-base models is successful even with simple averaging, while DeBERTa-large is hard to merge, which hints to an interaction between merge-ability and pre-training objective. We believe a more comprehensive study of such factors is an interesting direction of future work.

**Impact of Scaling Non-Diagonal Values in Inner Product Matrices**. We noticed when $\alpha = 1.0$ (*i.e.*, no scaling), RegMean yields degenerated performance on T5-base and DeBERTa when merging two models, while slightly decreasing $\alpha$ to 0.9 eliminates the issue. In the other extreme case when $\alpha = 0$, the inner product matrices become diagonal and RegMean simply reweigh rows of weight matrices, making the method similar to Simple Average. We plot the pairwise merging performance of RegMean with $0 \le \alpha \le 1$ in Figure 5a for T5-base and DeBERTa-large, as well as the performance of merging multiple T5 models in 5b. We observe that the performance of RegMean is mostly stable between $\alpha = 0.1$ and 0.9, but suddenly drops at $\alpha = 1.0$. When merging multiple T5-base models, both in-domain and OOD performs reaches maximum at $\alpha = 0.1$ and slowly drops with an increase in $\alpha$, whereas OOD performance suffers a slightly larger drop.

**Limitations.** We note that the requirement of inner product matrices in RegMean (and Fisher Information in Fisher-weighted averaging) can be a limitation. To merge existing models released online

Figure 6: Examples of improved out-of-domain generalization performance when incrementally merging a subset of individual models in the order of their OOD performance compared to merging all models. The main comparison is against the best individual model $f_{1..N}$ (shown in the dashed line).

without these statistics, a few training examples (see Appendix C for the sensitivity to the number of training examples) are needed to compute them. Besides, there is a risk that inner product matrices may reveal information about training data. Quantitatively measuring information leakage in these statistics should be a good direction of research in the area of privacy.

## 6 RELATED WORK

**Model Merging and Weight Averaging.** Past research studied model merging for different end goals. Izmailov et al. (2018); Gupta et al. (2020); Wortsman et al. (2022) aim to improve model performance by averaging weights across different checkpoints or different runs. Cha et al. (2021); Arpit et al. (2021); Ramé et al. (2022) study domain-generalization by averaging weights of models trained over the same datasets with different configurations. Matena & Raffel (2021) study merging using Fisher-weighted averaging with the aim of improving performance on a single target task by leveraging other 'donor' tasks. Choshen et al. (2022) show fusing fine-tuned models with simple weight-averaging creates a better starting point of fine-tuning for new tasks. Weight averaging was also used by Li et al. (2022) for building language models with multi-domain capabilities where new domain 'experts' are initialized using weight averaging from the existing experts. Wang et al. (2022) use weight averaging to fuse knowledge learned when training multiple adapters with the aim of obtaining better few-shot capabilities and increased model robustness. Merging updates of private models is a crucial intermediate step in **federated learning** (McMahan et al., 2017; Li et al., 2019). However, key in federated learning algorithms is that the joint model is iteratively updated in multiple rounds, which is not allowed for model merging. The success of simple arithmetic mean for model merging has been explained from the perspective of loss landscapes and linear mode connectivity (Frankle et al., 2020; Neyshabur et al., 2020; Draxler et al., 2018; Ainsworth et al., 2022). Further, improved merging algorithms aim to match permutations between the weights of different models (Singh & Jaggi, 2020; Nguyen et al., 2021; Ainsworth et al., 2022; Wang et al., 2020a), which is a complementary line of effort to our work. We experiment with permutation matching algorithms and present our analysis in Appendix D.

**Knowledge Fusing via Distillation.** Recent work has used the knowledge distillation framework to fuse the capabilities of multiple teacher models by distilling them into a smaller student model at fine-tuning or pre-training stage (Khanuja et al., 2021), albeit requiring full access to data for distillation. Dataless distillation, although for computer vision architectures and not using Transformer-based approaches, was attempted in (Lopes et al., 2017; Nayak et al., 2019). These have the additional disadvantage of not having a closed form solution and are thus not computationally efficient.

## 7 CONCLUSIONS AND FUTURE WORK

This paper studied the problem of fusing knowledge of multiple fine-tuned language models by model merging without access to training data. We proposed a new method inspired by linear models named Regression Mean (RegMean). We introduced a series of experimental setups in which we demonstrated that our method outperforms other alternatives to dataless merging or ensembling. Further, in non-i.i.d. and out-of-domain experiments, we showed that model merging can outperform individually trained models. Merged models are also very practical, especially when compared to hosting multiple models, as the merging algorithm is very efficient, adds a minimal number of additional parameters and has a similar inference speed to any individual model.

The implications of model merging are wide ranging from efficient intermediary-task selection to improve performance to combining models trained with private data in a federated learning setup. Future work can focus on merging models with different initialization or architectures, merging models sequentially at scale or merging pre-trained models before the fine-tuning stage.

ACKNOWLEDGMENTS

Xisen Jin is supported by a Bloomberg Data Science Ph.D. Fellowship.

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

# A    DERIVATION OF THE COMPLETE FORMULATION OF REGMEAN

Consider merging of $K$ linear models. We have the optimization problem formulation,

$$\min_{W} \quad \sum_{i}^{i\in\mathcal{K}} \|W^T X_i - W_i^T X_i\|^2 + \sum_{i}^{i\in\mathcal{K}} (W - W_i)^T \Lambda_i (W - W_i) \tag{3}$$

where for all $i$, $W, W_i \in \mathbb{R}^{m\times n}$, $X_i \in \mathbb{R}^{N_i \times m}$, and $\Lambda_i = \text{diag}(\lambda_{i1}, \lambda_{i2}, ..., \lambda_{iK}) \succeq 0$. The second term is a regularization term that encourages $W$ to be close to $W_i$, where $\lambda_{ij}$ is the regularization strength for $j$-th row of $W_i$. Here, $\lambda_{ij}$ can be set as any non-negative values. The optimal solution for this problem is,

$$W_M = [\sum_{i}^{i\in\mathcal{K}} (X_i^T X_i + \Lambda_i)]^{-1} \sum_{i}^{i\in\mathcal{K}} [(X_i^T X_i + \Lambda_i) W_i] \tag{4}$$

*Proof.* We compute the gradient of the objective function (noted as $L$) w.r.t the merged weight $W$.

$$\frac{\partial L}{\partial W} = \sum_{i}^{i\in\mathcal{K}} (-2X_i^T X_i W_i + 2X_i^T X_i W) + \sum_{i}^{i\in\mathcal{K}} (-2\Lambda W_i + 2\Lambda W) \tag{5}$$

We see $L$ is convex w.r.t. $W$. Therefore, we may find minizer of $L$ by letting $\frac{\partial L}{\partial W} = 0$.

$$\sum_{i}^{i\in\mathcal{K}} (X_i^T X_i W_i + \Lambda W_i) = \sum_{i}^{i\in\mathcal{K}} (X_i^T X_i + \Lambda) W^* \tag{6}$$

$$W^* = [\sum_{i}^{i\in\mathcal{K}} (X_i^T X_i + \Lambda_i)]^{-1} \sum_{i}^{i\in\mathcal{K}} [(X_i^T X_i + \Lambda_i) W_i] \tag{7}$$

$\square$

Usually, in linear regression, the regularization strength $\Lambda_i$ is manually specified as a constant value. However, in our case, the scale of $X_i^T X_i$ may differ a lot across models, layers, or datasets. Therefore, we let $\Lambda_i$ to scale with $X_i^T X_i$, and set $\Lambda_i = \gamma \, \text{diag}(X_i^T X_i)$, where $\gamma$ is a fixed scalar, so that,

$$W_M = [\sum_{i}^{i\in\mathcal{K}} (X_i^T X_i + \gamma \, \text{diag}(X_i^T X_i))]^{-1} \sum_{i}^{i\in\mathcal{K}} [(X_i^T X_i + \gamma \, \text{diag}(X_i^T X_i)) W_i] \tag{8}$$

This formulation is equivalent to increasing the scale of diagonal items of inner product matrices $X_i^T X_i$. Decreasing all non-diagonal items of inner product matrices by multiplying $\alpha = \frac{1}{1+\gamma}$ has the same effect, as we have done in Sec. 3.3.

$$W_M = [\sum_{i}^{i\in\mathcal{K}} (\frac{1}{1+\gamma} X_i^T X_i + \frac{\gamma}{1+\gamma} \, \text{diag}(X_i^T X_i))]^{-1} \sum_{i}^{i\in\mathcal{K}} [(\frac{1}{1+\gamma} X_i^T X_i + \frac{\gamma}{1+\gamma} \, \text{diag}(X_i^T X_i)) W_i] \tag{9}$$

## B  Details for Datasets, Preprocessing, Metrics, and Training

**GLUE.** For GLUE (Wang et al., 2018) experiments, we use CoLA (Warstadt et al., 2019), SST-2 (Socher et al., 2013), MRPC (Dolan & Brockett, 2005), STS-B (Cer et al., 2017), MNLI (Williams et al., 2018),QNLI (Rajpurkar et al., 2016), QQP, and RTE (Giampiccolo et al., 2007) datasets the GLUE task collections. We run evaluation on the official development sets because test labels are hidden. We compute Matthews Correlation for CoLA, Pearson Correlation for STS-B, and accuracy for all other tasks.

To study merging models trained on non-i.i.d. partitions, we construct two partitions for each of the GLUE tasks. We first randomly sample a "key class" from the task and draw 80% of data of the class from the training set and put them into one partition. The rest of the data constitute the other partition. We uniformly draw examples that do not belong to the "key class" from one partition to the other so that two partitions have the same number of examples. We uniformly sub-sample each partition so that each partition has 1,000 training examples.

**Emotion.** For emotion classification, we use the preprocessed datasets by Oberländer & Klinger (2018). We use DailyDialogs (Li et al., 2017), CrowdFlower, TEC (Mohammad, 2012), Tales-Emotion (Alm et al., 2005), and ISEAR (Scherer & Wallbott, 1994) for training domain-specific models. We use Emoint (Mohammad & Bravo-Marquez, 2017), SSEC (Schuff et al., 2017), ElectoralTweets (Mohammad et al., 2015), GroundedEmotions (Liu et al., 2017), and AffectiveText (Strapparava & Mihalcea, 2007) as held-out datasets for evaluating out-of-domain generalization. All the selected datasets have the classes *anger, disgust, fear, joy, sadness, surprise* in their label space, while some of them have more classes (*e.g.* guilt). For in-domain performance of each dataset, we compute Macro-F1 of all classes that present in the dataset. For out-of-domain performance, we only compute Macro-F1 over *anger, disgust, fear, joy, sadness, surprise*. In some of the datasets, inputs may be associated with multiple emotion labels. We therefore formulate the emotion classification task as a multi-label classification task for all datasets. Table 4 summarizes statistics of the datasets.

|  | Train | Dev | Test |
|---|---|---|---|
| *In-domain* | | | |
| DialyDialog | 72,085 | 10,298 | 20,596 |
| CrowdFlower | 27,818 | 3,974 | 7,948 |
| TEC | 14,735 | 2,105 | 4,211 |
| Tales-Emotion | 10,339 | 1,477 | 2,955 |
| ISEAR | 5,366 | 766 | 1,534 |
| *Out-of-domain* | | | |
| Emoint | | | 7,102 |
| SSEC | | | 4,868 |
| ElectoralTweets | | | 4,056 |
| GroundedEmotions | | | 2,585 |
| AffectiveText | | | 1,250 |

Table 4: Statistics of emotion classification datasets.

On RoBERTa and DeBERTa, we create a binary classification head for each class. We exclude the classification heads that are not learned in the training process when merging the weights of classification heads – *e.g.* if one dataset has the class "guilt" but the other does not, the weights of the classification head for "guilt" of the other model will not be used for merging.

For T5, we reformulate the task into a sequence-to-sequence format with the template: *does the sentence express {class_name}? {sentence}.* with possible outputs *yes* or *no*. Such an example will be created for each class that present in the dataset. During evaluation, we treat the exact match *yes* as the the prediction of the positive label, and otherwise treat as prediction of the negative label.

|  | Train | Dev | Test |
|---|---|---|---|
| *In-domain* | | | |
| OntoNotes:bc | 12,719 | 2,269 | 2,355 |
| OntoNotes:bn | 13,233 | 1,598 | 1,666 |
| OntoNotes:mz | 7,793 | 729 | 877 |
| OntoNotes:nw | 40,466 | 6,778 | 2,702 |
| OntoNotes:tc | 13,162 | 1,671 | 1,403 |
| OntoNotes:wb | 39,140 | 5,117 | 5,103 |
| *Out-of-domain* | | | |
| CoNLL | | | 3,684 |
| Twitter | | | 2,395 |

Table 5: Statistics of NER datasets.

**NER.** We use 6 domains (newswire, broadcast news, broadcast conversation, magazine, telephone conversation and web data) in OntoNotes (Hovy et al., 2006) for training 6 domain-specific individual models. For testing out-of-domain generalization, we use CoNLL Sang & De Meulder (2003) and a Twitter NER data set Rijhwani & Preotiuc-Pietro (2020). Table 5 summarizes statistics of the datasets.

|  | DailyDialog | CrowdFlower | TEC | Tales-Emotion | ISEAR |
|---|---|---|---|---|---|
| DailyDialog | 8.22 | 13.95 | 16.68 | 13.38 | 14.69 |
| CrowdFlower |  | 22.11 | 28.52 | 22.78 | 26.26 |
| TEC |  |  | 29.90 | 26.55 | **31.21** |
| Tales-Emotion |  |  |  | 18.28 | 24.19 |
| ISEAR |  |  |  |  | 30.06 |

Table 7: OOD performance when merging two RoBERTa-base emotion classification models (with same head initialization) with RegMean. Diagonal items represent OOD performance of individual models. We show OOD performance is dependent on the models used for merging.

**Implementation.** We use huggingface's transformer library (Wolf et al., 2019) to download pre-trained LM checkpoints and fine-tune the models. We specifically note that we use the forward function hook feature in PyTorch (Paszke et al., 2019) to obtain the inputs of all linear layers in order to compute inner product matrices. It makes the code implementation of RegMean agnostic to the model architecture.

**Training Details.** We fine-tune DistilBERT-base, RoBERTa-base, and DeBERTa-large with an initial learning rate 1e-5, and fine-tune T5-base with an initial learning rate 1e-4. We use AdamW optimizer throughout the experiments. The learning rate gradually warms up in the first 6% of training steps and linearly decay to 0. We train models with a batch size of 16 and for 10 epochs on GLUE, 30 epochs on emotion classification and 20 epochs on NER. We evaluate the performance of the model after each epoch and resume the best performing checkpoint at the end of training.

| $N$ | In-domain | OOD |
|---|---|---|
| 1 | 37.42 | 20.55 |
| 10 | 40.09 | 22.09 |
| 100 | **40.62** | 22.64 |
| 1,000 | 38.73 | 22.61 |
| 5,000 | 40.56 | **22.78** |

Table 6: Enumerating different setups of $N$ (numbers of batches of size 16 for computing inner product matrices) in merging all five RoBERTa-base models fine-tuned on emotion classification datasets. We report average performance over in-domain and out-of-domain (OOD) datasets.

## C  SENSITIVITY ANALYSIS

**Number of batches for computing inner product matrices.** In our main experiments, we use $N = 1,000$ batches (of size 16) for computing inner product matrices. We present additional analysis about the effect of $N$ and summarize results in Table 6. In general, performance improves as we increase $N$, but the performance soon saturates around $N = 100$.

**Alternative methods for regularization.** As we mentioned in Sec. 3.3 and Appendix A, we reduce non-diagonal items of inner product matrices by a fixed scale $\alpha$, which has a regularization effect of encouraging merged weights to be closer to individual model weights. Here we present analysis of an alternative regularization method, which adds a fixed scalar $\beta$ to diagonal items instead of relatively scaling them.

We experiment with emotion classification on T5 where regularization seems to be most necessary. We merge each pair of models on 5 emotion classification datasets and report the average performance over all pairs (a setting similar to Figure. 3) in Table 8. We see relative scaling achieves clearly better performance

|  | In-domain F1 |
|---|---|
| *Adding a constant to diagonals* |  |
| $\beta = 0.01$ | 28.24 |
| $\beta = 0.1$ | 33.74 |
| $\beta = 0.2$ | 39.13 |
| $\beta = 0.5$ | 34.70 |
| *Relative scaling of non-diagonals* |  |
| $\alpha = 0.1$ | **40.32** |

Table 8: Comparison of performing regularization by adding a constant to diagonals or relative scaling of non-diagonals of inner product matrices. We merge T5-base Emotion Classification models and evaluate average in-domain F1.

than adding a constant to diagonals. As we mentioned in Appendix A, this may be caused by differences in the scale of inputs in different layers, models, and datasets, which makes it difficult to find a single additive regularizer.

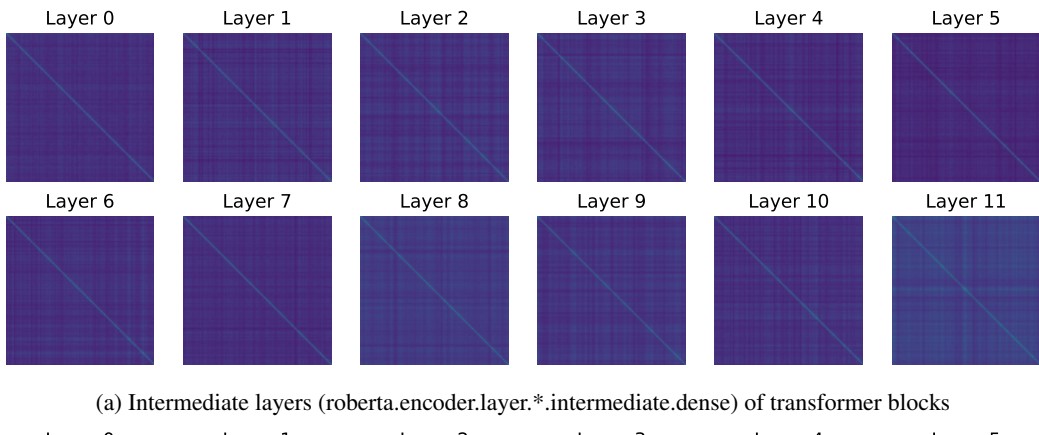

(a) Intermediate layers (roberta.encoder.layer.*.intermediate.dense) of transformer blocks

(b) Output layers (roberta.encoder.layer.*.output.dense) of transformer blocks

Figure 7: Visualizing $\ell^2$ distance between pairs of $n$ weight vectors in $W_A \in \mathbb{R}^{m \times n}$ and $W_B \in \mathbb{R}^{m \times n}$. Smaller values are highlighted in the heatmaps. We fine-tune RoBERTa-base models on two different emotion classification datasets. The resulting matrix $T$ is used as ground metrics for computing optimal transport in weight-based matching in (Singh & Jaggi, 2020).

**Choice of models to merge and its effect on OOD performance.** Table 7 summarizes OOD performance when merging each pair of RoBETa-base emotion classification models with same head initialization with RegMean. We see the OOD performance is clearly dependent on the models chosen for merging. Merging TEC and ISEAR models, which correspond to two individual models that achieve best OOD performance, produces a model that achieves best OOD performance.

## D  PERMUTATION MATCHING ALGORITHMS FOR MERGING LANGUAGE MODELS

Several existing works (Singh & Jaggi, 2020; Ainsworth et al., 2022) propose algorithms to match weight permutations in two models before merging, as models with similar outputs may involve distinct permutations in their weights. However, experiments in these works do not cover transformers LMs. In this section, we present an analysis to address two research questions about permutation matching algorithms in the setup of merging language models fine-tuned from shared pretrained weights: (1) does the issue of weight permutation exist in this setup? (2) do existing permutation matching algorithms improve the performance of model merging?

We experiment with merging two RoBERTa-base models fine-tuned on emotion classification datasets. We visualize results on merging models trained on Tales-Emotion and ISEAR in Figures 7 and 8.

**Weight-Based Matching.** We apply weight-based matching in OTFusion (Singh & Jaggi, 2020). To find permutations between weight matrices $W_A$ and $W_B$ in the same layer of two different models, the algorithm computes a ground metrics matrix $M \in \mathbb{R}^{n \times n}$, where $n$ is the dimension of the output. Each element $M_{ij} \in M$ measures $\ell^2$ distance between a pair of weight vectors $W_A^{:,i}$ and $W_B^{:,j}$.

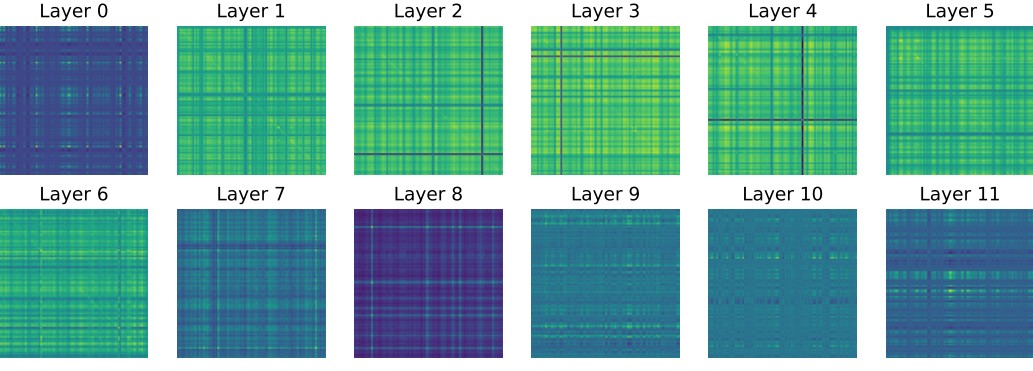

(a) Intermediate layers (roberta.encoder.layer.*.intermediate.dense) of transformer blocks

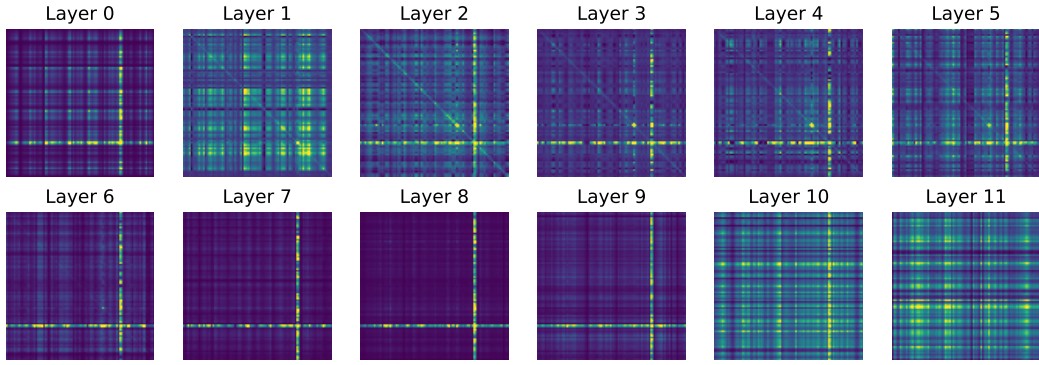

(b) Output layers (roberta.encoder.layer.*.output.dense) of transformer blocks

Figure 8: Visualizing $Z_A^T Z_B$, where $Z_A = \text{gelu}(W_A^T X_A)$ and $Z_B = \text{gelu}(W_B^T X_B)$ are the activations of the layers. We fine-tune RoBERTa-base models on two different emotion classification datasets. The resulting $Z_A^T Z_B$ is used for computing activation-based matching in (Ainsworth et al., 2022)

Assuming no permutations in weights, we should expect the diagonal items of $M$ (distance of weight vectors in the corresponding positions) to be much smaller than non-diagonal items. Otherwise, we may obtain non-trivial permutations by solving an optimal transport problem with $M$.

In Figure 7, we visualize the matrix $M$ on the two-layer MLP after each transformer block, which is the only place where linear layers are stacked without residual connections in transformers, making weight permutations most likely to happen. However, in Figure 7, we see a clear picture that the diagonal items of $M$ are significantly smaller than non-diagonals. The results imply there is no permutations in weights. In this case, the permutation matrix we obtain by solving optimal transport is a trivial identity matrix.

We conjecture that sharing the same pretrained LM weight initialization contributes to stability in training, resulting in no permutations in weights. The residual connections in transforms may further prevent weights in other modules from getting permuted.

**Activation-Based Matching.** We apply activation-based matching in Git Re-Basin (Ainsworth et al., 2022). The algorithms relies on a similarity matrix $C \in \mathbb{R}^{n \times n}$ that measures pairwise similarity of activations over $N$ training examples in a certain layer. More formally, $C$ is computed as $Z_A^T Z_B$, where $Z_A, Z_B \in \mathbb{R}^{N \times n}$ are activations at a given layer in the models $f_A$ and $f_B$. The algorithm solves a linear assignment problem with $C$ to obtain permutations in activations. Similarity, if there is no permutation, we expect the diagonal items of $C$ to be large.

We visualize the matrix $C$ in Figure 8. We see a different picture from weight-based matching that $C$ is far from being diagonal. This allows activation-based matching algorithms to produce non-trivial permutation matrices. However, as we apply these permutations, we obtain performance that is far below simple average without matching. We conjecture that in our setup permutations of activations could not faithfully represent permutations in weights. Though we just present empirical findings in this paper, we consider figuring out the reasons for such discrepancy as an interesting future work.

