# OpenReview forum: "Dataless Knowledge Fusion by Merging Weights of Language Models"
_ICLR.cc/2023/Conference — ICLR 2023 poster_

### Official Review · Reviewer_PCeq · 2022-10-25

**Confidence:** 3
**Correctness:** 3
**Technical Novelty And Significance:** 2
**Empirical Novelty And Significance:** 3
**Recommendation:** 6

**Clarity, Quality, Novelty And Reproducibility:**

The paper is mostly quite clear, aside from the important omissions discussed above.  A couple of other minor questions:

Sec 5.1 – “different label distributions,” how different are they?

A sentence that says what the domains are from the different-domains experiment (e.g. a couple of examples of the domains) would be helpful.

Regarding novelty, I feel that the approach is novel in detail although the high-level approach seems quite similar to Fisher-weighted averaging.

**Strength And Weaknesses:**

Strengths

This paper presents an elegant technique and presents it clearly.  It includes a number of helpful experiments that show that the technique performs well, albeit not uniformly better than all baselines.

Weaknesses

The paper fails to explain some important relationships between its approach and previous work, and may be missing important comparisons.  I also had a question about the ensemble baseline and I think comparing against other ensemble alternatives may be important.  I discuss these in more detail below.

Section 3.2 says it will explain the relationship between the proposed method and Fisher-weighted averaging, but it doesn’t really (it only discusses a distinction between an extreme special case of the proposed method and Fisher, and only very briefly and abstractly).  More discussion of how the proposed approach differs from Fisher and why we would expect it to perform better in certain situations is really critical.

In the experiments, the paper says the ensemble works “by obtaining logits from all models and doing an argmax”—this is confusing.  Ensembles I’m familiar with tend to average the probability outputs of each model, and then take the argmax of that average.  You could also average the logits (effectively, multiplying the probability outputs).  But taking the max over all the models, which is what this text seems to be saying, is not something I’ve seen.  Maybe I’m just unfamiliar with this approach and/or maybe it works better on these data sets than the other possibilities (especially in this setting with non-iid partitions), but the paper needs to explain this one way or another.

Regarding potentially missing baselines, the last sentence of the model merging paragraph in the related work section mentions several “improved merging algorithms”---how do these differ from the proposed method?  Why does the paper not include an experimental comparison against the best of these?

**Summary Of The Paper:**

This paper presents a simple and well-motivated technique for merging different models in a "dataless" way, and evaluates the method in comparison to simple averaging and Fisher-weighted averaging.

**Summary Of The Review:**

While this paper presents an elegant and non-trivially effective technique on an important problem, I was not convinced that it explained or evaluated its relationship to previous work well enough for publication.  However, if other reviewers (especially ones more directly familiar with the recent previous work) are enthusiastic about the work, I would not try to stand in the way of acceptance.

---

> ### Author Response · Authors · 2022-11-15
> **Response to Reviewer PCeq**
>
> We thank the reviewer for insightful comments. We see most of the questions are related to comparison to baselines. Here are our responses to individual questions.
>
> **Q1. More discussion of how the proposed approach differs from Fisher and why we would expect it to perform better in certain situations is really critical.**
>
> While Fisher information also has a formal interpretation, it relies on an assumption that the posterior distribution of model parameters (given training data) is gaussian. This assumption can be unrealistic which may result in inferior performance.
>
> In contrast, RegMean does not rely on such assumption. For linear models, we have shown that RegMean produces an optimal solution that minimizes $\ell^2$ distance in predictions.
>
> **Q2. The approach is novel in detail although the high-level approach seems quite similar to Fisher-weighted averaging**
>
> While RegMean and Fisher-weighted averaging have a seemingly similar procedure of pre-computing certain statistics (Fisher Information or inner product matrices), the formulations of merging are very different, and they rely fundamentally on different underlying information (Fisher vs inner product). We also see a significant difference in empirical results between Fisher-weighted averaging and RegMean.
>
> **Q3. Regarding potentially missing baselines, the last sentence of the model merging paragraph in the related work section mentions several “improved merging algorithms”---how do these differ from the proposed method?**
>
> We thank the reviewer for bringing up the issue. We present analysis of weight matching algorithms in our General Response to Reviewers in a separate comment. To summarize,
>
> - Weight-based matching in [1] suggests no permutations in the weights of two fine-tuned language models
> - Activation-based matching in [2] indicates permutations in activations, but the performance is lower than Simple Average as we align the weights following the matching
> - RegMean and permutation matching algorithms are orthogonal and complementary lines of efforts to the model merging problem.
>
> We refer the reviewer to the General Response (in a separate thread) for more details. We would appreciate it if the reviewer could check the thread.
>
> **Q4. In the experiments, the paper says the ensemble works “by obtaining logits from all models and doing an argmax”—this is confusing.**
>
> We are sorry about this clarity issue. We indeed averaged logits of individual models before doing an argmax, as what the reviewer has mentioned. We have updated descriptions about model ensembling in Sec. 4.2.
>
> **Q5: Sec 5.1 – “different label distributions,” how different are they?**
>
> We included how the non-IID partitions were created in Appendix A due to the space limit. In the new version, we added pointers to Appendix in the main text.
>
> We create non-IID partitions as follows: we first randomly sample a “key class” from the task and draw 80% of data of the class from the training set and put them into one partition. The rest of the data constitutes the other partition.
>
> **Q6: While this paper presents an elegant and non-trivially effective technique on an important problem, I was not convinced that it explained or evaluated its relationship to previous work well enough.**
>
> We thank the reviewer for appreciating the contribution of our paper. We hope our responses have addressed your concerns about comparisons to baselines and are happy to answer further questions.
>
> References:
>
> [1] Sidak Pal Singh and Martin Jaggi. Model fusion via optimal transport. Advances in Neural Information Processing Systems, 33:22045–22055, 2020.
>
> [2] Samuel K Ainsworth, Jonathan Hayase, and Siddhartha Srinivasa. Git re-basin: Merging models modulo permutation symmetries. arXiv preprint arXiv:2209.04836, 2022.

---

> ### Author Response · Authors · 2022-11-21
> **Response to Reviewer PCeq**
>
> We thank the reviewer again for reviewing our paper. Our responses and new experiments try to address the reviewer’s concerns about choices and details about baselines. We specifically present analysis about permutation matching algorithms in a separate thread. We are looking forward to your responses and are happy to have further discussions.

---

### Official Review · Reviewer_hdqV · 2022-10-25

**Confidence:** 4
**Correctness:** 4
**Technical Novelty And Significance:** 4
**Empirical Novelty And Significance:** 3
**Recommendation:** 8

**Clarity, Quality, Novelty And Reproducibility:**

This paper is very clear and novel with high quality from my perspective.


**Strength And Weaknesses:**

### Strengths:

1. I think the model merging problem studied in this paper is a very important direction to pursue – it is an efficient way of fusing knowledge from multiple large models without training or increasing the inference cost.
2. The proposed method RegMean is derived clearly from linear regression, and is novel and simple.
3. The experiments are comprehensive including multiple challenging settings and detailed analysis over pretrained models and hyperparameters. Particularly, the paper tests merging more than 2 models which the Fisher merging paper did not test. I think the insights revealed by these results could be useful for future researchers.
4. The experimental results of RegMean are good, demonstrating the usefulness of model merging in various settings and outperforming recent baselines like Fisher merging.


### Weaknesses:

1. I think one limitation of RegMean is that it requires access to the Gram matrix stats of the training data of each individual model. Even though this is much more reasonable than accessing the original training data, it would still make RegMean not applicable in many cases. This problem exists for Fisher merging as well, and average merging is the most flexible
2. I would like to see a more detailed discussion on other baselines mentioned in the related work by "improved merging algorithms" and why the authors do not compare with them.
3. I feel some results from this paper are contradictory with the ones in Matena et al. (the Fisher merging paper) – this paper observed performance drop in all pairwise test settings (Figure 3, Figure 4), while Matena et al. observe gains over task-specific models when merging donor tasks with the target task. Particularly, Table A2 in Matena et al.  and Figure 4 in this paper are both on the GLUE tasks but with different conclusions, in Matena et al there is no clear performance drop. I wonder how the authors explain this difference, is it because the pretrained models are different (Bert v.s. DistillBert/Roberta)? The performance drop in pairwise merging seems pretty consistent in this paper though.



### Questions:
In Table 2, are the results a mixture of test examples from multiple domains? If so, how are domain-specific numbers calculated?


```
Update during rebuttal:

I agree with the authors that the permutation lines of work are orthogonal and complementary to this paper, and the added discussion on this aspect is helpful. I also appreciate that the authors clarify my confusions in the response. Therefore, I increase my score to 8.
```


**Summary Of The Paper:**

This paper focuses on merging individual models on the parameter space without access to the respective training data. The individual models could come from different domains or even different tasks, their knowledge could be fused by merging without any training, and the resulting merging model could perform relatively well on all domains/tasks. Notably, the inference cost after merging is the same as the individual models. The authors propose the Regression Mean (RegMean) method to merge model parameters derived from merging linear models. RegMean uses the Gram matrices of the training data to reweight and linearly combine rows in weight matrices during merging. This paper consists of comprehensive experiments to examine merging multiple models trained on different domains / different tasks and test on in-domain and out-of-domain settings. Empirical results demonstrate the effectiveness of the method compared with recent baselines.

**Summary Of The Review:**

I recommend acceptance of this paper since it studies an important and intriguing problem – model parameter merging, proposes new algorithms, and achieves good results in demonstrated comprehensive experiments. The weakness is that the method makes strong assumption on access to the Gram matrices of private training data.

---

> ### Author Response · Authors · 2022-11-15
> **Response to Reviewer hdqV**
>
> We thank the reviewer for the thoughtful comments. We respond to individual questions below.
>
> **Q1: Comparison with other merging algorithms (permutation matching algorithms) mentioned in the related works section**
>
> We performed additional experiments and analysis about permutation matching algorithms and included them in Appendix D. We also address the question in a separate thread, “General Response to Reviewers”.
>
> To summarize, we find that,
>
> - Weight-based matching in [1] suggests no permutations in the weights of two fine-tuned language models
> - Activation-based matching in [2] indicates permutations in activations, but the performance is lower than Simple Average as we align the weights following the matching
> - RegMean and permutation matching algorithms are orthogonal and complementary lines of efforts to the model merging problem.
>
> We would appreciate it if the reviewer could check our responses in the separate thread.
>
> **Q2: Some of the results contradict Matena et al. [3]**
>
> We hope to clarify that the differences in results stem from the differences in setups. [3] tries to improve performance over a single “target” dataset by merging weights from a “donor” model. The results of [3] are on a single dataset, while ours are an average of two or more datasets. In addition, [3] performs a grid search over coefficients of merging two models ($\alpha W_A + (1-\alpha) W_B$) that maximize performance over a validation set of the target dataset. Therefore, the merged model is at least as good as an individual model on the target dataset, because the coefficient of the “donor” model can be simply set as zero.
>
> Our paper, in contrast, tries to obtain a single model that performs well on all domains or tasks. We also do not perform grid search of coefficients so that the framework does not rely on a validation set. Besides, grid search may become intractably expensive when merging multiple models.
>
> **Q3: In Table 2, are the results a mixture of test examples from multiple domains? If so, how are domain-specific numbers calculated?**
>
> For domain-specific numbers, we evaluate each domain-specific individual model only over its (single) training domain and report the average.
>
> *We thank the reviewer for recognizing our contributions. We hope our responses have addressed your questions and are happy to answer further questions.*
>
>
> References:
>
> [1] Sidak Pal Singh and Martin Jaggi. Model fusion via optimal transport. Advances in Neural Information Processing Systems, 33:22045–22055, 2020.
>
> [2] Samuel K Ainsworth, Jonathan Hayase, and Siddhartha Srinivasa. Git re-basin: Merging models modulo permutation symmetries. arXiv preprint arXiv:2209.04836, 2022.
>
> [3] Michael Matena and Colin Raffel. Merging models with fisher-weighted averaging. arXiv preprint arXiv:2111.09832, 2021.

---

> > ### Comment · Reviewer_hdqV · 2022-11-17
> > **Thank you**
> >
> > Thanks for the reply! I have updated my review accordingly.

---

### Official Review · Reviewer_fiB5 · 2022-10-25

**Confidence:** 4
**Correctness:** 4
**Technical Novelty And Significance:** 4
**Empirical Novelty And Significance:** 4
**Recommendation:** 8

**Clarity, Quality, Novelty And Reproducibility:**

- To the best of my knowledge the proposed algorithm is novel and interesting.
- The paper is well written with clear description of the motivation, the algorithm, the experimental setup and the results.
- The paper uses open datasets and model architectures and appears to be easily reproducible.


**Strength And Weaknesses:**

Strengths:

- The algorithm is simple to understand, well justified, and as far as I can tell novel (though I am not that familiar with prior work for this specific task).
- The RegMean algorithm appears to work better than existing alternatives (simple averaging and Fisher averaging) while being roughly as efficient in terms of computation time and memory.
- The empirical results are well carried out with good baselines and comparisons for different models.

Weaknesses:
I have two concerns with this algorithm which I think should be addressed more prominently than they are. They are a bit buried late in the intro or the methodology sections.
- First, the algorithm is really only intended to work in the case where every model starts from the same pertained baseline. It's pretty clear the algorithm would not work even for the same model architecture trained on the same dataset with a different random initialization as there is no attempt to solve the permutation problem. I would recommend being more upfront with this limitation.
- Second, the primary justification for this approach is to maintain data privacy (since otherwise the models could all be trained on the data). However, the algorithm reveals the quite a bit of data for each task, roughly the size of the model which can be fairly large. I would be more upfront in discussing this weakness.
- The algorithm is only used for "linear" layers. It would be interesting if it the authors also examined how well it worked for other types of layers (the self-attention matrices, convolutional layers, etc.).


Other minor comments:

- "Margin Models Trained on Different Tasks": Point of experiment unclear.
- Results in Table 3 for Fisher under DeBERTa-large for Emotion with same/diff are very strange. Why does using a different head result in such a huge performance improvement compared to the same head?
- In Figure 6a does it make much difference which two models you choose? Or is performance just worse when you go from any two models to three and so on.
- Did you try experiments with using just a simple regularization (e.g. adding some small value to the diagonal of the Hessian prior to inversion)? I notice you scale the diagonal up in your algorithm but I'm curious if you tried the simpler/more traditional alternative and how it compared?
- In the computation of the input features for each layer's regression problem you use 1000 batches (section 3.4?). How important is this parameter? Is the performance of the algorithm quite sensitive to it or not that much?

Nits:

- The Gram matrix is usually the matrix of sample size x sample size - the matrices that come up in the normal equations e.g. X^T X aren't usually called Gram matrices. I would avoid using this name as it is confusing.
- In section 3.4, 1k means 1,000? I would just write it out for clarity.
- Unneeded period after "Figure. 5a" at the bottom of page 8.


**Summary Of The Paper:**

This paper proposes a new strategy for merging the parameters of models which have been fine-tuned on different datasets. The merging algorithm is intended for models which are all fine-tuned from the same base model. The primary contribution of the paper is the algorithm for merging models. The algorithm is formulated as a least squares problem where the goal is to minimize the squared distance between the merged model parameter and the individual model parameters applied to the individual datasets. The authors perform experiments on emotion classification and named entity recognition with a variety of models. They show that their approach improves over alternative prior algorithms though is generally not as good as full multi-task training.

**Summary Of The Review:**

The proposed algorithm is novel and an improved solution to the problem the authors describe. The algorithm is also fairly simple to understand and efficient to implement which are good features. The authors also show support that RegMean works well by demonstrating results on two NLP tasks with several models.

My main concern are the two primary limitations (as discussed above). These limitations might make the situations in which this algorithm is applicable somewhat niche.

---

> ### Author Response · Authors · 2022-11-15
> **Response to Reviewer fiB5**
>
> We thank the reviewer for the insightful comments! Here are our responses to individual questions.
>
> **Q1: The algorithm is really only intended to work in the case where every model starts from the same pertained baseline.**
>
> RegMean assumes models share the same pretrained initialization, which is similar to the limited past work on this topic. We will feature this assumption more prominently. We employ this assumption because of the prevalent practice of fine-tuning models from a released pretrained model checkpoint in NLP. We believe using different initializations is out of scope for this initial work on the topic of merging pretrained language models, but is a relevant area for future research.
>
> However, we note RegMean can be built on top of permutation matching algorithms naturally in case such permutation exists. We updated our experiments to include permutation matching algorithms (in General Response & Appendix D). We include more discussions in our general response to reviewers and would appreciate it if the reviewer could check our detailed response to this question.
>
> **Q2: Data privacy is a motivation but inner product matrices may still leak information and are large in size**
>
> This is similar to the case of the Fisher-weight averaging baseline, where the Fisher Information matrix is shared. We believe quantitatively measuring information leakage in these statistics (and even in model parameter themselves)  to be a good direction of research in the area of privacy. We added a paragraph to discuss this limitation and motivate future research in Sec. 5.3.
>
>
> We agree with the reviewer that data privacy is just one of the motivations. Another major benefit of model merging is efficiency in fusing knowledge in multiple models with a closed-form solution.
>
> **Q3: The algorithm is only used for Linear layers. What is the effect on other types of layers, e.g. convolution, and self-attention matrices?**
>
> We focus on transformers language models where the majority of the layers are linear layers. This includes attention layers, where the only learnable parameters are linear transformation layers of the inputs. Extension to convolutional layers can be an interesting future work.
>
> **Q4: In Figure 6a (incrementally merging more models) does it make much difference which two models you choose? Or is performance just worse when you go from any two models to three and so on.**
>
> We show that the performance is dependent on which models to merge. We include additional experiments in Appendix C. When merging two individual models that achieves best OOD performance, we obtain even better OOD performance of 31.21. In contrast, when merging two worst individual models, we only obtain OOD performance of 13.38.
>
> **Q5: Can we use simple regularization (adding small value to the diagonal of XTX prior to inversion) instead of proportionally scaling non-diagonal items?**
>
> We present additional experiments in Appendix C. The performance of adding a value to the diagonal is lower than our current practice of scaling diagonals proportionally. As we mentioned in Appendix A, this may be caused by the differences in the scale of XTX matrices across datasets, layers, and models.
>
> **Q6: How important is the number of batches used for computing inner product matrices?**
>
> We present an additional set of experiments in Appendix C, where we enumerate different numbers of batches for computing inner product matrices. In general, performance improves as we increase n, but the performance soon saturates.
>
> **Q7: XTX is not called gram matrix in convention. Gram matrix usually has the dimension input_num * input_num**
>
> We thank the reviewer for bringing up this issue. To avoid confusion, we have changed all such mentions to “inner product matrices”.

---

### Official Review · Reviewer_MwUi · 2022-10-26

**Confidence:** 4
**Correctness:** 3
**Technical Novelty And Significance:** 3
**Empirical Novelty And Significance:** 2
**Recommendation:** 5

**Clarity, Quality, Novelty And Reproducibility:**

Clarity, Quality: The paper is well written and easy to follow.
Novelty: The paper has its merit but still some claims need to be further validated. The technical novelty is somewhat limited. The problem setting is limited to a newly proposed one and the proposed method is not validated in the existing settings.
Reproducibility: The authors provide psuedo code and implementation description. The code release plan is unknown.

**Strength And Weaknesses:**

Strength
1. The proposed setting is practical and has its merits in the real world.
2. The proposed method has clear motivation and derivation.
3. The authors evaluate the proposed method in several settings including in-domain and out-of-domain,  showing improvements compared to  various baseline.


Weakness
1. One of the motivation is that sharing training data may have data privacy issue. A detailed analysis regarding risks about sharing of  Gram matrices along with the models is needed.
2. The proposed setting is more or less too specific. To protect the training data privacy, the federated learning serves the similar purpose. The authors did not compare their models to this setting. Another qualified baseline like ensemble is not included in the experimental Table 1.
3. The improvements are not significant compared to Fisher weight merging on GLUE benchmark.
3. The problem setting is limited to that the training data is not allowed to share. It is also helpful to understand if this method is able to bring improvements when training data is allowed to share. Such a setting is more commonly used when studying weight merging.
4.  The performance improvements brought by weight merging may decay as the training data size increases. The training data is limited to 1000 in  Table 1 without a clear reasons. To better understand the effectiveness of the proposed mechanism,  it is important to study if the weight merging could bring improvements when the training data is in a relatively large scale.

**Summary Of The Paper:**

The authors work on merge  individual models built on different training data sets to obtain a single model. More specifically, the authors propose a dataless knowledge fusion which is guided by weights that minimize prediction differences between the merged model and the individual models.

**Summary Of The Review:**

The authors propose a dataless knowledge fusion based on weight merging. The motivation of the proposed method is clear. However, the studies setting comes with many restrictions and generalization ability of this method is somewhat limited. More details can be found in Strength And Weaknesses section.

---

> ### Author Response · Authors · 2022-11-15
> **Response to Reviewer MwUi**
>
> We thank the reviewer for the thoughtful comments. Here are our responses to individual questions.
>
> **Q1: The gram matrix (inner product matrix) may leak information about training data**
>
> This is similar to the case of the Fisher-weight averaging baseline, where the Fisher Information matrix is shared. We believe quantitatively measuring information leakage in these statistics (and even in model parameter themselves) to be a good direction of research in the area of privacy. We added a paragraph to discuss this limitation and motivate future research in Sec. 5.3.
>
> **Q2: The proposed setting is more or less too specific. To protect the training data privacy, Federated learning serves a similar purpose.**
>
> We believe that model merging represents a general problem. Model merging is a crucial step in Federated Learning; while it also applies to other scenarios. Federated learning studies a scenario with multiple rounds of “syncing” among individual models, while we study weight merging in a single round (see Figure 1 in the paper). This makes model merging applicable, for example, for merging models that are already released online (with fisher information, inner product matrices, or a few training data points), where we cannot perform multiple rounds of syncing with agents that released models.
>
>
> **Q3: The improvements are not significant compared to Fisher weight merging on the GLUE benchmark.**
>
> We believe the reviewer refers to our experiments of merging models trained on non-IID partitions of GLUE (beginning of Sec. 5.1) where RegMean performs similarly to Fisher, but not worse. On our experiments of merging models trained on different GLUE tasks (end of Sec. 5.1), the improvements are significant.
>
> We actually included non-IID partitions as a proof-of-concept experiment, or a sanity check, where simple averaging could improve performance over individual models. On more challenging setups, the improvements of RegMean are much more significant.
>
>
> **Q4: The problem setting is limited to that the training data is not allowed to be shared. It is also helpful to understand if this method is able to bring improvements when training data is allowed to be shared.**
>
> We hope to clarify the practical use case of the “dataless” setup. (1) as what the reviewer has mentioned, it applies to merging models released online where data cannot be shared. (2) the algorithm requires no training, making the algorithm very computationally efficient for fusing knowledge in large numbers of models and model combinations.
>
> **Q5: The performance improvements brought by weight merging may decay as the training data size increases. The training data is limited to 1000 in Table 1 without a clear reason. To better understand the effectiveness of the proposed mechanism, it is important to study if the weight merging could bring improvements when the training data is on a relatively large scale.**
>
> Table 1 presents a proof-of-concept experiment of merging models trained on non-IID partitions. Our intention here is to create a simple setup to validate the benefit of model merging algorithms where even simple averaging could improve over individual models. The rest of experiments are performed in realistic setups where models are trained on full datasets from different domains or tasks. In this new version we include tables of data statistics Appendix B. We show that a number of datasets contain tens of thousands of training examples.
>
> *We hope our responses have addressed the concerns of the reviewer and we are happy to answer any further questions.*

---

> ### Author Response · Authors · 2022-11-21
> **Response to Reviewer MwUi**
>
> We thank the reviewer again for reviewing our paper. In our responses, we try to show model merging represents an important and practical problem and dissect its differences from federated learning. We also highlight the improvement of our algorithms over baselines in many challenging setups presented in Sec. 5. We also updated the paper to discuss the limitations introduced by inner product matrices as suggested by the reviewer.
>
> We are looking forward to your responses and are happy to have further discussions.

---

### Author Response · Authors · 2022-11-15
**General Response (1/3)**

We thank all the reviewers for their thoughtful questions, comments and recognition of contributions of this work. We believe efficiently fusing knowledge of multiple large models without training is an important research problem. We propose a novel and simple merging algorithm, RegMean, and systematically evaluate it in a variety of setups, where it clearly improves over comparators.

Here is a summary of main updates we made to the paper.
- Comparison and discussions of **permutation matching algorithms**: Appendix D.  We also respond in this thread below *[Reviewers fiB5, hdqV, PCeq]*
- Requirements of inner product (Gram) matrices: we discuss potential limitation and motivate future research at the end of Sec. 5.3 *[Reviewers MwUi, fiB5, hdqV]*
- Effect of models selected for merging to out-of-domain performance: Appendix C *[Reviewer fiB5]*
- Effect of number of examples used for computing inner product matrices: Appendix C *[Reviewer fiB5]*
- Alternative regularization techniques in RegMean: Appendix C *[Reviewer fiB5]*
- We changed all mentions of "Gram matrices" to "inner product matrices" to avoid confusion *[Reviewer fiB5]*

Below, we address common questions from the reviewers.

---

> ### Author Response · Authors · 2022-11-15
> **General Response (2/3)**
>
> **Q: Discussions about permutation matching algorithms for weight averaging.**
>
> We present additional experiments and have added discussions in Appendix D. We show that in our setup of merging two fine-tuned LMs (details below):
>
> - Weight-based matching in [1] suggests **no permutations in the weights** of two fine-tuned language models
> - Activation-based matching in [2] indicates **permutations in activations**, but the **performance is lower** than Simple Average as we align the weights following the matching
> - RegMean and permutation matching algorithms are **orthogonal and complementary** lines of research to the model merging problem.
>
>
> **1. Weight-based matching**
>
> We apply weight-based matching in OTFusion [1]. To summarize the algorithm, it computes distance between each pair of weight vectors in $W_A$ and $W_B$, resulting in a $n\times n$ “ground metrics” matrix $M$ (where n is the dimension of the output). Assuming no permutations in inputs or weights, we may expect the scale of the diagonals of $M$ (i.e. weight vectors in the same position) to be significantly smaller than non-diagonals. Otherwise, we may figure out permutations by computing optimal transport with M.
>
> We present our analysis on the two-layer MLP layers after each transformer block, the only place where two linear layers are stacked without residual connections in transformers, where permutations are most likely to happen. We visualize the ground metrics matrix $M$ in Figure 7 in Appendix D, as we merge two RoBERTa-base models trained on different datasets. We see a clear pattern in Figure 7 that the diagonals of $M$ are significantly smaller than non-diagonals, suggesting no permutations. In this case, the resulting merged model is the same Simple Averaging, as optimal transport is simply an identity matrix.
>
> We conjecture that sharing the same pretrained LM weight initialization contributes to stability in training, resulting in no permutations in weights. The residual connections in transforms could further prevent weights in other modules from getting permuted.
>
> **2. Activation-based matching**
>
> We apply activation based matching which was introduced in a concurrent work [2]. The matching is computed by solving a linear assignment problem over $n\times n$ pairwise similarity matrix of activations of two models in the same layer, more formally $C=Z_A^TZ_B$, where $Z_A$, $Z_B$ are the activations over a joint set of training examples of two datasets, and $n$ is the dimension of the activation. Similarly, if there is no permutation, we expect the diagonal items of $C$ to be large.
>
> We present a similar analysis as weight-based matching in Figure 8. The figure clearly indicates there are permutations: $C$ is far from diagonal, and diagonals are barely highlighted in only some layers. This allows activation-based matching to produce non-trivial permutation matrices. However, when we apply these permutations to weights, we see the performance is far below Simple average.
>
> We conjecture that in our setup of merging LMs, permutations of activations could not faithfully represent permutations in weights, as we see such contradicting results.
>
> Though we just presented empirical findings, we consider understanding the reasons for such discrepancy as a relevant future work in permutation matching methods.
>
> **3. Finally, we want to highlight that RegMean and permutation matching (as above) methods are orthogonal and complementary lines of research.**
>
> RegMean tries to obtain a single multi-task model that performs well on all tasks or domains. *Figuring out the perfect permutation matching, or having two models without any permutation in the first place, does not guarantee the merged model to be an optimal multi-task model*. This can be seen from the example of optimally merging two linear models (which has no permutations in weights) in the Sec 3 and how simple average misses the optimal solution.
>
> Meanwhile, just as Reviewer fiB5 mentioned, *RegMean does not handle weight permutation*. This can be seen from the formulation of RegMean (Eq. 1). When weights are permuted, we are not likely to obtain a good W that minimizes the output distance from both individual models.
>
> Fortunately, there is a simple way to combine RegMean and permutation matching algorithms. We may first transform the other model $f_B$ to another model $f_B’$ that is equivalent modulo permutation to $f_B$ and matches $f_A$. We then apply RegMean to merge $f_A$ and $f_B’$.
>
> However, in our current experiment setups, we see either no permutations in weights or low performance of matching algorithms. We believe combining RegMean and permutation matching and figuring out the setup where such a combination is necessary is an exciting future work.

---

> > ### Author Response · Authors · 2022-11-15
> > **General Response (3/3)**
> >
> > **Q: Plan for code release**
> >
> > We will release code upon acceptance.
> >
> > References:
> >
> > [1] Sidak Pal Singh and Martin Jaggi. Model fusion via optimal transport. Advances in Neural Information Processing Systems, 33:22045–22055, 2020.
> >
> > [2] Samuel K Ainsworth, Jonathan Hayase, and Siddhartha Srinivasa. Git re-basin: Merging models modulo permutation symmetries. arXiv preprint arXiv:2209.04836, 2022.

---

### Decision · Program_Chairs · 2023-01-20

**Decision:**

Accept: poster

**Justification For Why Not Higher Score:**

Privacy concerns of sharing data set inner product matrices could be more prominent.


**Justification For Why Not Lower Score:**

Clarity, exhaustive experiments, simplicity/practicality of the method.


**Metareview: Summary, Strengths And Weaknesses:**

Summary: This paper proposes ensembling language models by parameter fusion. The method applies to cases where the different models originated from fine tuning the same model on different datasets. The method aims to maximize the agreement between the merged predictor and individual models, while not requiring access to individual fine tuning datasets for fusion. It is easy to implement and compares favorably to alternatives.

Strength:
- clarity of the paper, strong motivation and related work section, exhaustive experiments.
- simplicity of the method.
- empirical performance over alternatives on the IID GLUE setup.
- the authors addressed reviewer comments with care (Section 5.3, appendix B, appendix C...)

Weaknesses:
- the method requires sharing an inner product matrix, which might leak information about the dataset. The defining "dataless" merging might be discussed (the authors added 5.3 to highlight this weakness).
- the method requires the merged model to be fine tuned from the same checkpoint (which is still a common setting in application though).


**Note From Pc:**

if the above contains the word "oral" or "spotlight" please see: "oral" presentation means -> notable-top-5% and "spotlight" means -> notable-top-25%. As stated in our emails, we are disassociating presentation type from AC recommendations